# Genetic Heterogeneity and Mutated PreS Analysis of Duck Hepatitis B Virus Recently Isolated from Ducks and Geese in China

**DOI:** 10.3390/ani13081282

**Published:** 2023-04-08

**Authors:** Shuqi Xu, Xinhao Mu, Xin Xu, Congying Bi, Jun Ji, Yunchao Kan, Lunguang Yao, Yingzuo Bi, Qingmei Xie

**Affiliations:** 1Henan Provincial Engineering Laboratory of Insects Bio-reactor, Henan Provincial Engineering and Technology Center of Health Products for Livestock and Poultry, Nanyang Normal University, Nanyang 473061, China; 2Henan Provincial Engineering and Technology Center of Animal Disease Diagnosis and Integrated Control, Nanyang Normal University, Nanyang 473061, China; 3College of Animal Science, South China Agricultural University, Guangzhou 510642, China

**Keywords:** duck hepatitis B virus, phylogenetic analysis, complete genome sequence, recombination, infection

## Abstract

**Simple Summary:**

The duck hepatitis B virus (DHBV), owing to its high similarity to the human hepatitis B virus (HBV), is an ideal model for HBV research. In this study, 23 complete genomes of Chinese strains of DHBV were analyzed using phylogenetic methods, and their recombinant events were predicted and analyzed. These data aid in understanding the intergenotypic recombination of HBV and are expected to promote further research on the epidemiology and evolution of DHBV.

**Abstract:**

In this study, we detected 12 duck and 11 goose flocks that were positive for duck hepatitis B virus (DHBV) using polymerase chain reaction and isolated 23 strains between 2020 and 2022 in China. The complete genomes of goose strains E200801 and E210501 shared the highest identity (99.9%), whereas those of strains Y220217 and E210526 shared the lowest identity (91.39%). The phylogenetic tree constructed based on the genome sequences of these strains and reference strains was classified into three major clusters: the Chinese branch DHBV-I, the Chinese branch DHBV-II, and the Western branch DHBV-III. Furthermore, the duck-origin strain Y200122 was clustered into a separate branch and was predicted to be a recombinant strain derived from DHBV-M32990 (belonging to the Chinese branch DHBV-I) and Y220201 (belonging to the Chinese branch DHBV-II). Additionally, preS protein analysis of the 23 DHBV strains revealed extensive mutation sites, almost half of which were of duck origin. All goose-origin DHBV contained the mutation site G133E, which is related to increased viral pathogenicity. These data are expected to promote further research on the epidemiology and evolution of DHBV. Continuing DHBV surveillance in poultry will enhance the understanding of the evolution of HBV.

## 1. Introduction

Duck hepatitis B virus (DHBV) was first discovered in Peking ducks in 1980 [1] and subsequently reported in Germany and other countries worldwide [2,3,4,5]. DHBV is a member of the genus *Avihepadnavirus* belonging to the family Hepadnaviridae [6]. Its genome has complete minus and incomplete plus strands and is approximately 3.0 kb in length [6]. The genome is maintained in the circular conformation via a short cohesive overlap between the two DNA strands. Its genome organization includes at least three partially overlapping open reading frames (ORFs) that encode the core protein (ORF-C), the viral polymerase (ORF-P), and the surface protein antigens known as PreS/S proteins (ORF-S) [7].

Hepatitis B virus (HBV) and DHBV share similar characteristics in various aspects of genetic organization, virus replication, and viral life cycle [8,9,10]. The main features of hepadnaviral infection were first discovered using the DHBV model and were subsequently confirmed in HBV and other hepadnaviruses [11]. Therefore, DHBV serves as an animal infection model of human HBV and has been widely used for comparative studies [6].

In HBV, different genotypes are related to clinical progression, response to antiviral treatment, and prognosis, which can influence the clinical outcome of HBV infection [12,13]. However, research on the molecular patterns of intergenotypic recombination of HBV remains limited. Studying the recombination of other hepadnaviruses, especially DHBV, could provide key information for understanding HBV recombination [14].

In both DHBV and HBV, the PreS protein can induce a high titer of neutralizing antibodies, thus promoting virus variation under the host’s immune pressure [9,15,16]. Therefore, the preS-encoded sequence is highly variable and has been suggested to be involved in the species-specific recognition of target cells [17,18]. Unlike HBV, DHBV does not cause severe clinical disease in ducks. However, DHBVs that contain the G133E strain mutation in the preS protein can cause acute liver injury even in the absence of severe inflammation [9]. The vertical route is the major transmission mode for DHBV, and the induced persistent infection in breeding may reduce egg hatching and duckling growth [19,20]. A recent report showed that glucose homeostasis is dysregulated in ducks infected with DHBV [21]. These reports imply that economic losses to the global poultry industry caused by DHBV infection should not be ignored.

In this study, we examined the molecular epidemiological characteristics of DHBV infection in Central and Eastern China. The strains’ phylogenetic characteristics and the mutation of PreS amino acids were also analyzed.

## 2. Materials and Methods

### 2.1. Sample Collection, Virus Screening, and Isolation

Between 2020 and 2022, 540 serum samples were randomly obtained from 23 duck and 31 goose farms (10 samples per farm) located in the Henan, Anhui, Jiangsu, and Hubei provinces of China. The serum samples were obtained from the jugular vein blood using aseptic techniques and centrifuged at 5000 rpm for 10 min. Then, 0.2 mL of the supernatant was collected for nucleic acid extraction using an extraction kit (EasyPure Viral DNA/RNA Kit; TransGen Biotechnology, Inc., Beijing, China) as per the manufacturer’s instructions. The purity and concentration of the DNA/RNA samples were determined using biological spectrophotometry, and the samples were stored at −20 °C until use.

Based on the multiple alignments of the complete genome of DHBV available in GenBank, primer pairs (P1: 5′-GCCCAAATCTC(A/T)CCACA-3′ and P2: 5′-GGCAGAGGAGGAAGTCAT-3′; P3: 5′-AATTACACCCCTCTCCTTCGGAGC-3′ and P4: 5′-TAATTCTTAAGTTCCACATAGCC-3′) were designed to detect and amplify the complete genome sequence of DHBV using Primer Premier 5 software.

DHBV detection was performed in a 20-μL volume containing 1 μM of each primer (P1 and P2) and Max HS polymerase chain reaction (PCR) Master Mix containing a hot-start-modified DNA polymerase kit (TaKaRa Biotechnology Co., Ltd., Dalian, China). The PCR conditions included initial denaturation at 95 °C for 2 min, followed by 30 cycles at 95 °C for 30 s, 53 °C for 30 s, and 72 °C for 30 s, and a final extension step at 72 °C for 10 min. The detection results were determined via electrophoresis on a 1% agarose gel. The positive infection rate (%) was calculated as follows: positive infection rate of samples (%) = (number of positive samples/total number of samples) × 100. The positive infection rate of farms (%) was calculated as follows: (number of farms with positive infection rate/total number of farms) × 100.

The positive serum samples were collected and passed through a 0.22-μM filter after being screened using PCR with primers (P1 and P2). Then, 9-day-old duck embryo eggs were inoculated into the allantoic cavity with the filtered suspension (100 μL/embryo) and cultured in an incubator at 37 °C. The allantoic fluid was harvested 5 days after inoculation and evaluated using semiquantitative PCR with 20 cycles (as described above) for five serial passages. The intensity ratios of the PCR products under ultraviolet transillumination between each allantoic fluid sample and the standard DNA template (pDHBV) were calculated to determine the changes in the viral load [22].

### 2.2. PCR Amplification and Sequencing

The PCR amplification of the complete genome was performed in a 20-μL reaction mixture, which included a template DNA (>100 ng/L, extracted from the 100-μL allantoic fluid described above and the corresponding serum sample, respectively), 6-pmol P3/P4 primers, 1-μL of dNTP mix (2.5 mM each), 0.5-μL PrimerSTAR HS DNA polymerase (5 U/μL) (TaKaRa Biotechnology Co., Ltd., Dalian, China), and a supporting 10× reaction buffer. The amplification procedure was as follows: initial denaturation at 94 °C for 5 min, 30 cycles of denaturation at 94 °C for 30 s, annealing at 57 °C for 30 s, extension at 72 °C for 3 min and 10 s, and a final extension step at 72 °C for 10 min. The amplified products were cloned into a pMD18-T simple vector (TaKaRa Biotechnology Co., Ltd., Dalian, China) for classical dideoxy Sanger sequencing (HONGXUN, Suzhou, China).

### 2.3. Sequence Alignment and Mutation Analysis

The alignment of the whole genome sequence of the 23 DHBV strains and the reference strains was performed using MegAlign in DNAStar. Information on the reference strains is provided in Appendix A. Identity analysis of the complete genome between the obtained and reference strains was visualized with heatmaps using OriginPro 8.0 software (OriginLab Corporation, Northampton, MA, USA).

To visually reflect the difference in preS protein between the three variant DHBV strains (Y210928, Y200106, and E200422) and the classical reference strain 31, the SwissModel (https://swissmodel.expasy.org/interactive) (accessed on 20 February 2023) was used to predict the tertiary structure. The tertiary structure of the amino acid residues obtained from the SWISS-Prot database was presented using the PyMol 4.6.0 software.

### 2.4. Evolution and Recombination Analysis

The genetic evolution of the DHBV strains was constructed using the maximum likelihood method with Molecular Evolutionary Genetics Analysis (MEGA X) software with 1000 bootstrap replications [23]. Then, it was visualized using the online web server Evolview (http://www.evolgenius.info/evolview/) (accessed on 16 February 2023) [24].

To study the possibility of intergenic recombination, 55 complete DHBV genomes of the obtained and reference strains were analyzed to find evidence for possible intergenic recombination of DHBV. Potential parental sequences were identified using the Recombination Detection Program (RDP) version 4.83 with various methods, including GENECONV, BOOTSCAN, MaxChi, CHIMAERA, SISCAN, and 3SEQ [25].

## 3. Results

### 3.1. Sample Screening and Genome Amplification

In this study, 54 waterfowl farms were investigated: 4 farms out of 13 were positive for DHBV in Jiangsu Province (2 duck farms and 2 goose farms), 6 out of 15 in Henan Province (4 duck farms and 2 goose farms), 6 out of 11 in Hubei Province (2 duck farms and 4 goose farms), and 7 out of 15 in Anhui Province (4 duck farms and 3 goose farms). The results showed that the overall positivity rates for DHBV in duck and goose flocks were 52.2% (12/23) and 35.5% (11/31), respectively, which indicate its high prevalence in Chinese duck flocks. Using the PCR assay, DHBV was consistently detected in the allantoic fluids of the first inoculation of each sample. No obvious differences in the viral load were detected for any of the strains.

### 3.2. DNA Alignment and Identity Analysis

The final genome sequences of the 23 obtained DHBV strains contained 3024 bp or 3027 bp. Identity based on the sequences from the allantoic fluid of the first inoculation and the corresponding serum sample of each strain reached 100%. Of the 23 DHBV strains, 4 of the 12 duck strains were 3027 bp in length, and the remainder were 3024 bp in length; all goose strains were 3027 bp in length (Appendix A). The 23 DHBV strains exhibited 91.39% (duck strain Y220217 and goose strain E210526)–99.9% (goose strains E200801 and E210501) identity based on the sequence alignment of the entire genome. Of the 23 obtained DHBV strains and the 31 reference strains, the goose strains E210321 and E210526 exhibited the lowest similarity (88.22%) with the CH6 strain, whereas the duck strains Y220201 and GD3 exhibited the highest similarity (98.94%). The identity comparison of the genome sequences of the DHBV strains is shown in Figure 1.

The preS protein of the 23 DHBV strains contained extensive mutation, insertion, and deletion sites (Table 1). The Y200106, Y201009, and Y220217 strains contained sequential mutations between amino acid residues 67 and 114 in the PreS protein. Five strains contained the G133E mutation in the PreS protein and accounted for 41.67% (5/12) of the duck strains. The same mutation site was harbored in the goose strains and accounted for 100% (11/11) of the goose strains. Meanwhile, goose-origin DHBV strains contained specific mutation sites located at positions 185, 211, and 231.

### 3.3. Prediction of the Tertiary Structures of the Key Mutation Loci

The tertiary structures of three representative variant strains and strain 31 were modeled using the SwissModel homology modeling server. Compared with strain 31, Y210928, Y200106, and E200422 contained several mutations. The 133 (G-E) mutations changed the structure of the random crimp. Moreover, the three insertions located at loci 96, 148, and 164 remarkably altered the distribution of six major alpha helices (Figure 2).

### 3.4. Phylogenetic Analysis and Frequency Distribution

The distribution of the 23 DHBV strains and the 31 reference strains in the phylogenetic tree of the complete genome is shown in Figure 3. According to the constructed phylogenetic tree, the DHBV strains were separated into two major groups, the Western strain (DHBV-III) and the Chinese strain (DHBV-I and DHBV-II), with three main branches. Strains isolated from ducks and geese in this study were mostly clustered for the Chinese strains of DHBV-II and DHBV-III. Meanwhile, the duck-origin strain Y200122 and the reference strain SGHBV1-13 isolated from snow goose formed independent branches.

### 3.5. Recombination Analysis

In this study, the RDP4 and Simplot software tools were used to evaluate the complete genomes of the 23 DHBV strains and the 31 reference strains to investigate the likelihood of recombination. An intergenotypic recombinant strain, Y200122, was predicted to be recombined from DHBV-M32990 and Y220201 with high confidence based on the recombination examination methods in RDP4 (Table 2). The bootscan analysis used to detect recombination and estimate breakpoints within the parent strains is displayed in Figure 4.

## 4. Discussion

In this study, 23 of 54 farms in 4 Chinese provinces were found to be DHBV positive. In our previous report, we detected DHBV in serum and tissue samples and observed that the positivity rate in goose samples was slightly lower than that in duck samples [21,26]; we found the same to be true in this study. The investigation results indicated the high prevalence of DHBV in both duck and goose flocks in China.

The 11 goose-origin DHBV strains were merged with the duck-origin DHBVs isolated in this study, which belonged to 2 Chinese branches (DHBV-I and -II), and the reference strains clustered into 3 major genotypes corresponding to the previously identified “Chinese” and “Western” branches [16]. The strains obtained in this study belonged to the Chinese branches. Consistent with previous reports, the CH4, CH6, SD-02, and DHBV-XY strains belonged to the “Western” branch but were isolated in China, and DHBV-AJ006350 belonged to the “Chinese” branch but was isolated in Australia. These findings suggest that the definitions of branches based on regions are inaccurate [5]. Meanwhile, the snow goose HBV strains (SGHBV1-13) clustered in one branch showed genetic differences from the DHBV strains isolated from the geese tested in this study. Furthermore, the duck-origin strain Y200122 formed an independent evolutionary branch and was identified as a highly variable or recombinant strain.

Considering the evolutionary pattern of various viruses, an important way a virus can increase its genetic variety is via recombination between distinct genotypes. This recombination is thought to be crucial for the evolution of some viruses, especially intergenotypic recombination. Intergenotypic recombination has been reported for different viruses, including HIV-1 [27], hepatitis C virus [28], and more complexes in HBV [29,30]. Owing to their different genomic configurations (RNA or DNA, single-stranded or double- stranded) and different replication schemes, intergenotypic recombination of these viruses employs different molecular mechanisms [27,28,30]. In this study, the Y200122 strain was predicted to be a recombinant from DHBV-M32990 (belonging to the Chinese branch DHBV-I) and Y220201 (belonging to the Chinese branch DHBV-II). The locations of the recombinant DNA fragment and strain types produced were entirely different from those of the previously described Chinese CH4 and CH6 strains [14]. The CH4 and CH6 strains belong to the “Western” branch and are recombinants from DHBV16 (minor parent strain, belonging to the “Western” branch, DHBV-III) and GD3 (major parent strain, belonging to the Chinese branch, DHBV-II). Based on recombination characterization of DHBV in Central China from 2017 to 2019, strains Y190303HN and Y170101HB are recombinants from Chinese isolates Y180813HB (Chinese branch, DHBV-I) and E170101AH (Chinese branch, DHBV-II) with the Western isolate DHBV-XY (Western branch, DHBV-III) [26]. These findings indicate the complicated evolutionary trend and recombination pattern of DHBV.

No obvious clinical symptoms caused by DHBV infection were observed in the ducks and geese, including birds infected with strains containing the G133E mutation in the PreS protein, which has previously been reported to be related to an increase in pathogenicity, resulting in acute liver injury [9]. Similarly, no obvious difference in virus-load from allantoic fluid was noted between the strains containing the G133E mutation and those without the mutation. To determine the discrepancy caused by duck/goose species-specific differences in synergistic reaction related to other mutations, deep reverse genetic investigations and larger animal inoculation experiments are needed. Meanwhile, the goose-origin DHBV strains obtained in this study harbored these specific mutation sites as well as other mutation sites located at positions 185, 211, and 231. Considering that the preS protein has been proposed to be involved in species-specific recognition of target cells, the relationship between these specific mutation sites in PreS and host selection differences in host adaption needs to be explored further [8,9,17]. As described previously, the PreS protein in the DHBV strains isolated from ducks and geese in this study contained extensive mutations [18]. To the best of our knowledge, no DHBV vaccines are available in China. As the PreS protein can induce a high titer of neutralizing antibodies [31], it can be speculated that variations in this protein were likely due to immune pressure resulting from the wide circulation and persistent infection of this virus in China.

## 5. Conclusions

This study describes the high prevalence of DHBV in both ducks and geese in some provinces of China and highlights the benefits of understanding its prevalence and evolution. More importantly, the intergenotypic recombination and the highly variable PreS protein can be used as references for studying HBV.

## Figures and Tables

**Figure 1 animals-13-01282-f001:**
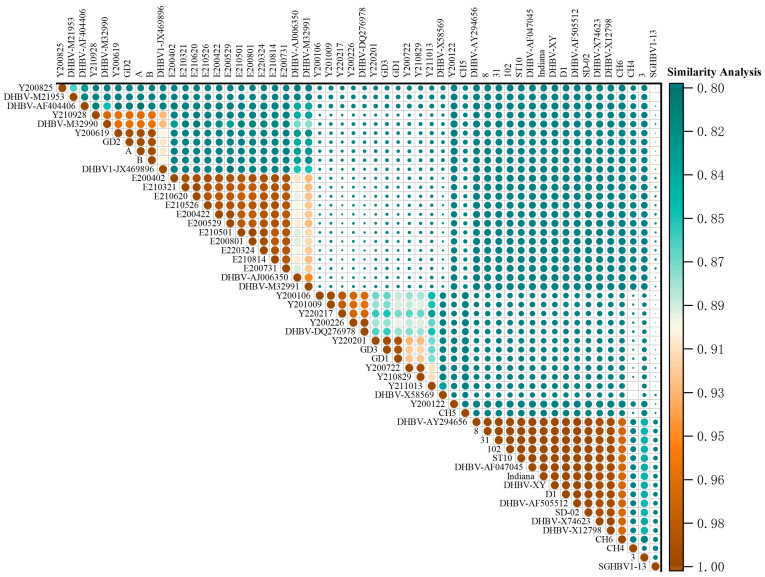
Heatmap of the similarity analysis of the entire genome of the DHBV strains and reference strains obtained in this study (upper right: different identity values are expressed in gradient colors ranging from 80% to 100%).

**Figure 2 animals-13-01282-f002:**
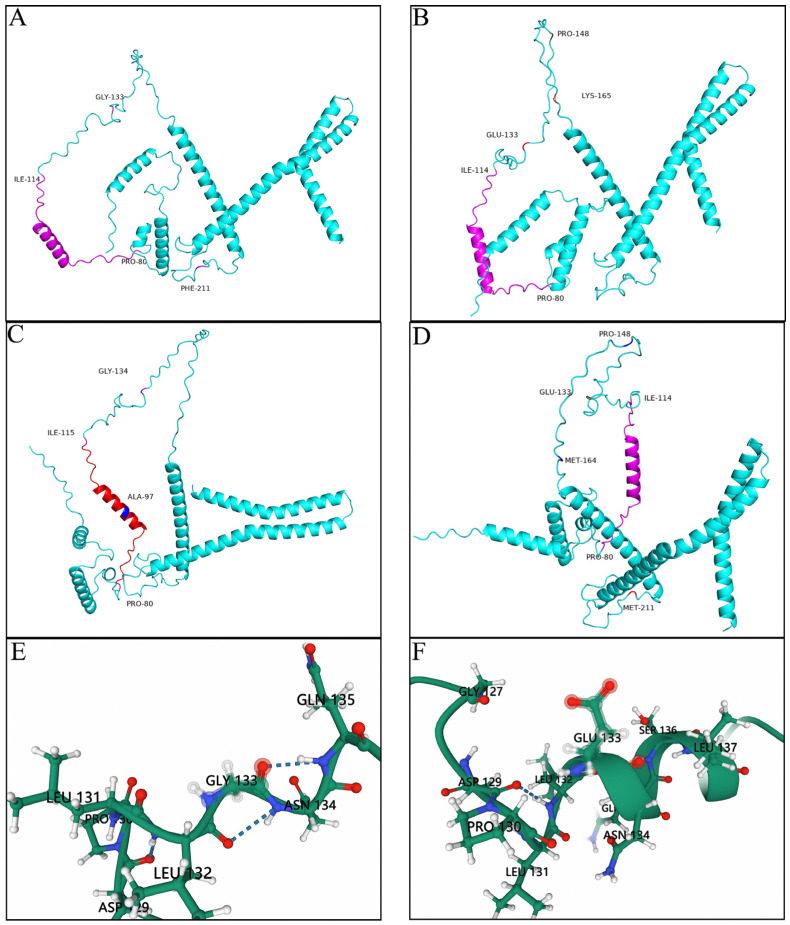
Cartoon scheme view of the preS protein structure. (**A**) preS protein structure of strain 31. (**B**) preS protein structure of strain Y210928. (**C**) preS protein structure of strain Y200106. (**D**) preS protein structure of strain E200422. (**E**) Conserved regional structure of GLY133 site of the preS protein. (**F**) Mutated regional structure of GLu133 site of the preS protein.

**Figure 3 animals-13-01282-f003:**
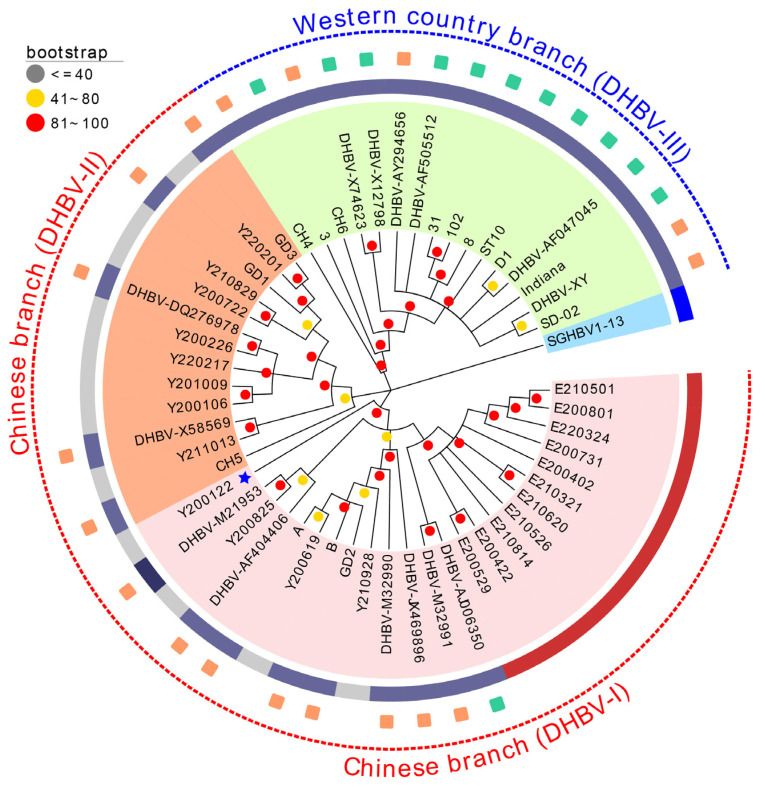
The DHBV phylogenetic tree based on the entire genome. In maximum likelihood analysis using MEGA X, the filled circles along the branches represent bootstrap values. Blue and red dotted lines represent the foreign and Chinese strains, respectively. Green and orange squares represent the Western and Chinese reference strains, respectively. The dark blue and gray solid rings represent the reference and research strains, respectively. Blue pentagrams denote recombinant strains.

**Figure 4 animals-13-01282-f004:**
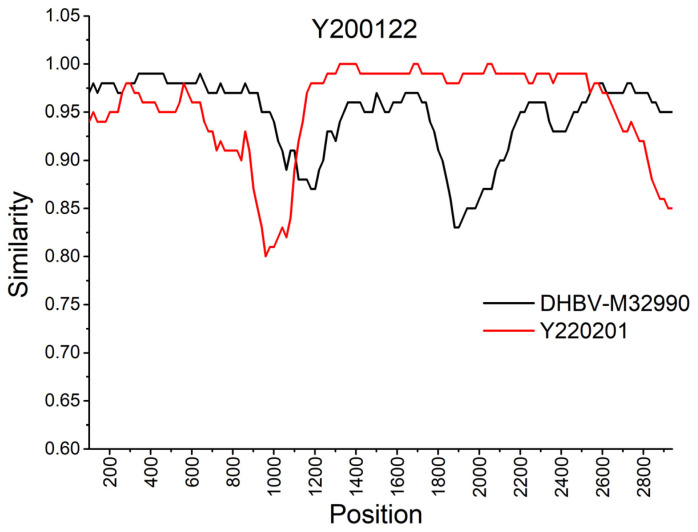
Recombination occurrence in the Y200122 strain analyzed using the Simplot software.

**Table 1 animals-13-01282-t001:** (A). Mutation sites of PreS compared with the amino acid residues of the DHBV reference strains (31, GenBank accession number: AY250901, 67–133 aa). (B). Mutation sites of PreS compared with the amino acid residues of the DHBV reference strains (31, GenBank accession number: AY250901,148–334 aa).

	**(A)**																																
Strains Name	Substitution of amino acid residues in PreS																																
	67	78	79	80	81	82	83	84	85	87	88	89	90	91	92	93	94	96 I ^a^	99	100	101	102	103	104	105	106	107	108	110	111	113	114	133
31	A	P	T	P	Q	E	I	P	Q	Q	W	T	P	E	E	D	Q		E	A	F	R	R	Y	Q	E	E	R	P	E	T	T	G
Y200825	- ^b^	-	A	-	-	-	-	-	-	-	-	-	-	-	-	-	-		-	-	-	-	-	-	-	-	-	-	-	-	-	-	E
Y200106	E	L	L	L	K	R	S	L	S	S	G	L	R	K	K	I	R	L	L	S	V	V	T	R	K	K	D	H	E	T	P	P	-
Y201009	E	L	L	L	K	R	S	L	S	S	G	L	R	K	K	I	R	L	L	S	D	V	T	R	K	K	D	H	E	T	P	P	-
Y210928	-	-	-	-	H	-	T	-	-	-	-	-	-	-	-	-	-		-	-	-	-	-	-	-	-	-	-	-	-	-	-	E
Y220201	E	-	A	-	-	-	-	-	-	K	-	-	-	-	-	-	-		-	-	-	-	-	-	-	-	-	-	-	-	-	-	-
Y211013	E	-	A	-	-	-	-	-	-	K	-	-	-	-	-	-	-		-	-	-	-	-	-	-	-	-	-	-	-	-	-	-
Y220217	E	L	L	L	K	R	S	L	S	N	G	L	R	K	K	I	R	L	L	S	V	V	T	R	K	K	D	P	E	T	P	P	-
Y200122	-	-	-	-	H	-	T	-	-	-	-	-	-	-	-	-	-		-	-	-	-	-	-	-	-	-	-	-	-	-	-	-
Y200226	E	-	A	-	-	-	-	-	-	K	-	-	-	-	A	-	-		K	-	-	-	-	-	-	-	-	-	-	-	-	-	-
Y200619	-	-	-	-	-	-	T	-	-	-	-	-	-	-	-	-	-		-	-	-	-	-	-	-	-	-	-	-	-	-	-	E
Y200722	E	-	A	-	-	-	-	-	-	K	-	-	-	-	-	-	-		-	-	-	-	-	-	-	-	-	-	-	-	-	-	E
Y210829	E	-	A	-	-	-	-	-	-	K	-	-	-	-	-	-	-		-	-	-	-	-	-	-	-	-	-	-	-	-	-	E
E200402	-	-	A	-	-	-	T	-	-	-	-	-	-	-	-	-	-		-	-	-	-	-	-	-	-	-	-	-	-	-	-	E
E200422	-	-	A	-	-	-	T	-	-	-	-	-	-	-	-	-	-		-	-	-	-	-	-	-	-	-	-	T	-	-	-	E
E200731	-	-	A	-	-	-	T	-	-	-	-	-	-	-	-	-	-		D	-	-	-	-	-	-	-	-	-	-	-	-	-	E
E210501	T	-	A	-	H	-	T	-	-	-	-	-	-	-	-	-	-		-	-	-	-	-	-	-	-	-	-	-	-	-	-	E
E210321	-	-	A	-	-	-	T	-	-	-	-	-	-	-	-	-	-		-	-	-	-	-	-	-	-	-	-	-	-	-	-	E
E210526	-	-	A	-	-	-	T	-	-	-	-	-	-	-	-	-	-		-	-	-	-	-	-	-	-	-	-	-	-	-	-	E
E210620	-	-	A	-	-	-	T	-	-	-	-	-	-	-	-	-	-		-	-	-	-	-	-	-	-	-	-	-	-	-	-	E
E210814	-	-	-	-	-	-	T	-	-	-	-	-	-	-	-	-	-		-	-	-	-	-	-	-	-	-	-	-	-	-	-	E
E220324	-	-	A	-	-	-	T	-	-	-	-	-	-	-	-	-	-		-	-	-	-	-	-	-	-	-	-	-	-	-	-	E
E200529	-	-	A	-	-	-	T	-	-	-	-	-	-	-	-	-	-		-	-	-	-	-	-	-	-	-	-	-	-	-	-	E
E200801	I	-	A	-	H	-	T	-	-	-	-	-	-	-	-	-	-		-	-	-	-	-	-	-	-	-	-	-	-	-	-	E
		**(B)**																															
Strains Name		Substitution of amino acid residues in PreS																															
		148 I ^a^			164 I ^a^			169		185		211	224		229		231		267	269		277		301	304		305		332	333		334	
31		- ^b^			-			S		S		K	I		V		S		G	T		A		T	L		S		Y	K		S	
Y200825		P			-			-		-		-	-		A		-		-	-		-		M	-		L		F	-		N	
Y200106		-			-			-		-		-	-		A		-		-	-		-		M	-		L		-	-		-	
Y201009		-			-			-		-		-	-		A		-		-	-		-		M	-		L		-	-		-	
Y210928		P			K			P		-		-	-		-		-		M	I		E		M	-		L		-	-		N	
Y220201		P			-			-		-			T		-		-		-	-		-		M	-		L		-	-		-	
Y211013		T			-			A		-		-	-		-		-		-	-		-		M	-		L		F	-		N	
Y220217		-			-			-		-		-	T		A		-		-	-		-		M	-		L		-	-		N	
Y200122		P			-			-		-		-	T		-		-		-	-		-		M	-		L		-	-		-	
Y200226		T			-			-		-		-	T		-		-		-	-		-		M	-		L		-	-		-	
Y200619		P			K			P		-		-	-		-		-		M	I		E		M	-		L		-	-		N	
Y200722		T			-			A		-		-	T		-		-		-	-		-		M	-		L		F	-		N	
Y210829		T			-			A		-		-	T		-		-		-	-		-		M	-		L		F	-		N	
E200402		P			K			A		G		-	-		A		F		M	I		E		M	P		L		-	R		N	
E200422		P			K			-		-		-	-		A		F		M	I		E		M	-		L		-	R		N	
E200731		P			K			A		G		-	-		A		Y		M	I		E		M	-		L		-	-		N	
E210501		P			K			A		G		-	-		A		F		M	I		E		M	-		L		-	-		N	
E210321		P			K			A		G		E	-		A		F		M	I		E		M	-		L		-	-		N	
E210526		P			K			A		G		E	-		A		F		M	I		E		M	-		L		-	R		N	
E210620		P			K			A		G		E	-		A		F		M	I		E		M	-		L		-	-		N	
E210814		P			K			A		G		-	-		A		F		M	I		E		M	-		L		-	R		N	
E220324		P			K			A		G		E	-		A		F		M	I		E		M	-		L		-	-		N	
E200529		P			K			-		G		-	-		A		F		M	I		E		M	-		L		-	R		N	
E200801		P			K			A		G		-	-		A		F		M	I		E		M	-		L		-	-		N	

^a^ I, insertion site; ^b^ “-” site of obtained strain same as the reference strain.

**Table 2 animals-13-01282-t002:** Recombination events and the related average *p*-values calculated using different recombination detection methods.

Event	RDP	GENOCOV	BOOTSCAN	MaxChi	CHIMAERA	SISCAN	3Seq
I	6.90 × 10^−8^	2.08 × 10^−3^	1.40 × 10^−4^	2.28 × 10^−8^	4.92 × 10^−7^	1.12 × 10^−7^	6.13 × 10^−10^

## Data Availability

All data generated or analyzed during this study are included in this article. Datasets are deposited in a publicly accessible repository: Datasets generated for this study can be found in GenBank: https://www.ncbi.nlm.nih.gov/Genbank/ (accessed on 6 April 2023). Genbank accession numbers are mentioned in the Materials and Methods of the article.

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
