# Peer review of "Genetic Heterogeneity and Mutated PreS Analysis of Duck Hepatitis B Virus Recently Isolated from Ducks and Geese in China"

_animals, 2023, doi:10.3390/ani13081282_

Round 1
Reviewer 1 Report
Very interesting paper and quite well done.
Couple of questions for authors
Introduction line 58: "Liver injury due to absence of severe inflammation". I would presume severe liver injury would be accompanied by inflammation?
Materials and methods Line 94. The authors mentioned inoculating the viruses into embryonated duck eggs. No further mention of these experiments appear to be mentioned? It was also unclear if the sequence analysis was performed on serum obtained DHBV/GHBV infected animals or on the egg grown virus. If egg grown virus, were mutations generated in the eggs?
Results line 140: "15 of 12 strains"?? 15 of 23?
Given the reported pathogenic nature of the G133E mutants, did these mutants grow to similar titres in eggs?.
Are any measures being undertaken to identify and remove DHBV/GHBV positive flocks? A rapid ELISA could be used to quickly screen farms for positivity. Flocks could be reconstituted with negative animals.
Author Response
Response to Reviewer 1 Comments
Point 1: Introduction line 58: "Liver injury due to absence of severe inflammation". I would presume severe liver injury would be accompanied by inflammation?
Response 1: Thanks for your recognition. We have read the relative reference and revised the statements, the liver injury have not been due to the inflammation.
Point 2: Materials and methods Line 94. The authors mentioned inoculating the viruses into embryonated duck eggs. No further mention of these experiments appear to be mentioned? It was also unclear if the sequence analysis was performed on serum obtained DHBV/GHBV infected animals or on the egg grown virus. If egg grown virus, were mutations generated in the eggs?
Response 2: Thanks for your advice. We have added the experiments method and results for embryo inoculation. Sequence amplification were from DNAs extracted from allantoic fluid and the corresponding serum sample, respectively, and identity based on the sequences from the allantoic fluid of the first inoculation and the corresponding serum sample of each strain reached 100%, respectively.
Point 3: Results line 140: "15 of 12 strains"?? 15 of 23?
Response 3: Thanks for your advice. We have changed “15 of 12 strains” to “4 of 12 strains”.
Point 4:Given the reported pathogenic nature of the G133E mutants, did these mutants grow to similar titres in eggs?
Response 4: Thanks for your advice. We have added the PCR assay based on the intensity ratios of the PCR products under ultraviolet transillumination between each allantoic fluid sample and the standard DNA template (pDHBV) were calculated to determine the changes in the viral load. No obvious differences in the viral load were detected for any of the strains.
Point 5:Are any measures being undertaken to identify and remove DHBV/GHBV positive flocks? A rapid ELISA could be used to quickly screen farms for positivity. Flocks could be reconstituted with negative animals.
Response 5: Thanks for your advice. Your advice is very important for quickly screen farms for positivity. However, no commercial ELISA kit for DHBV detection is available and we have not developed the ELISA assay. Therefore, we used PCR assay instead and would established the ELISA assay for detection of DHBV.
Reviewer 2 Report
This manuscript describes molecular analysis of recent DHBVs isolated from ducks and geese in China, revealing sequence variability and inter-genotypic combination of DHBVs. The finding enriches understanding of molecular epidemiology of DHBVs.
Given that the authors published “Genome analysis and recombination characterization of duck hepatitis B virus isolated from ducks and geese in central China, 2017 to 2019” on Poultry Science in 2023, can the authors clarify the significance and overall impact of this study compared to the published article please?
The major limitation of this manuscript is that it was not linguistically and scientifically written with the “Chinglish” feature. Below are several examples:
Lines 247-250: The clinical observations did not support increased viral pathogenicity due to genetic mutations in the PreS protein. Thus, this discrepancy has to be discussed and the way of expression needs to be paraphrased.
Lines 256-259: Are DHBV vaccines available in China? What is the mechanism of DHBV evolution? How do the authors conclude that immune pressure contribute to viral variation?
Line 83: As four primers were mentioned in the Materials and Methods, it is not clear of what kind of PCR (single or duplex PCR) was performed in the section (Line 83 to 91).
Line 95: How many days did each passage last? What are the indicator for further passage. However, virus isolation data wee not mentioned in the Results section.
Table 1 can be deleted by incorporating into Figure 1 and Supplementary Table 1.
There is not footnote in Table 2 B.
What does the number in the Row “strains” refer to in Table 2 A. It is difficult to be understood.
Last but not least: This research required animal ethic approval. The AEC approval number was not included in the manuscript.
Author Response
Response to Reviewer 2 Comments
Point 1: This manuscript describes molecular analysis of recent DHBVs isolated from ducks and geese in China, revealing sequence variability and inter-genotypic combination of DHBVs. The finding enriches understanding of molecular epidemiology of DHBVs.
Given that the authors published “Genome analysis and recombination characterization of duck hepatitis B virus isolated from ducks and geese in central China, 2017 to 2019” on Poultry Science in 2023, can the authors clarify the significance and overall impact of this study compared to the published article please?
Response 1: Thanks for the comments. The paper published in Poultry Science was focused on recombination analysis and the strains were isolated from 2017 to 2019. Outbreak of COVID-19 delayed the writing and review of this submission. And this paper were mainly talked about the extensive mutation of preS protein of DHBV, and the strains were isolated in last three years. We have added the previous published paper as reference and discussed it in the discussion section.
Point 2: The major limitation of this manuscript is that it was not linguistically and scientifically written with the “Chinglish” feature. Below are several examples.
Response 2: Thanks for the comments. We have improved the language again by native editors in NES Editing Institution. Many thanks for your time and patience.
Point 3: Lines 247-250: The clinical observations did not support increased viral pathogenicity due to genetic mutations in the PreS protein. Thus, this discrepancy has to be discussed and the way of expression needs to be paraphrased.
Response 3: Thanks for the comments. We have added the relative cmparison in the last paragraph in the discussion section.
Point 4: Lines 256-259: Are DHBV vaccines available in China? What is the mechanism of DHBV evolution? How do the authors conclude that immune pressure contribute to viral variation?
Response 4: Thanks for the comments. No commercial vaccines were used against DHBV, we have revised the relative statements and speculated the immune pressure might cause by wide circulation and persist infection.
Point 5: Line 83: As four primers were mentioned in the Materials and Methods, it is not clear of what kind of PCR (single or duplex PCR) was performed in the section (Line 83 to 91).
Response 5: Thanks for the comments. We have marked the description of primer pairs used in the Materials and Methods to make it clear.
Point 6: Line 95: How many days did each passage last? What are the indicator for further passage. However, virus isolation data wee not mentioned in the Results section.
Response 6: Thanks for the comments. We have added the experiments protocol and results for embryo inoculation.
Point 7: Table 1 can be deleted by incorporating into Figure 1 and Supplementary Table 1.
Response 7: Thanks for the comments. We have deleted the previous Table 1 and incorporating it into Supplementary Table 1.
Point 8: There is not footnote in Table 2 B.
Response 8: Thanks for the comments. We have added the footnote in Table 2 B.
Point 9: What does the number in the Row “strains” refer to in Table 2 A. It is difficult to be understood.
Response 9: Thanks for the comments. We have revised the description in this Table.
Point 10: Last but not least: This research required animal ethic approval. The AEC approval number was not included in the manuscript..
Response 10: Thanks for the comments. The ethical approval have been listed in Institutional Review Board Statement above the reference section.
Round 2
Reviewer 2 Report
The authors have made significant progress towards the publication of the manuscript. I only have a few minor comments for improvement.
Line 28-29/ Line 262: Please keep the expression "whether the G133E mutation increased virus pathogenicity" consistent in the manuscript.
Line 145: Please detail the viral load results.
Line 275: Please add the reference regarding PreS-induced neutralizing antibody.
Line 282: Please use appropriate wording for "references".
Author Response
Point 1: Line 28-29/ Line 262: Please keep the expression "whether the G133E mutation increased virus pathogenicity" consistent in the manuscript.
Response 1: Thanks for the comments. We feel very sorry for the unclear statements in the abstract. We have revised the words to make it consistent with the statements in Line 262 in the discussion section.
Point 2: Line 145: Please detail the viral load results.
Response 2: Thanks for the comments. We have added the viral load in the results section. Many thanks for your time and patience.
Point 3: Line 275: Please add the reference regarding PreS-induced neutralizing antibody.
Response 3: Thanks for the comments. We have added “Enhanced magnitude and breadth of neutralizing humoral response to a DNA vaccine targeting the DHBV envelope protein delivered by in vivo electroporation” as the last reference.
Point 4: Line 282: Please use appropriate wording for "references".
Response 4: Thanks for the comments. We have changed “references” to “research model”.